# Effect of NaCl and EDDS on Heavy Metal Accumulation in *Kosteletzkya pentacarpos* in Polymetallic Polluted Soil

**DOI:** 10.3390/plants12081656

**Published:** 2023-04-14

**Authors:** Mingxi Zhou, Zahar Kiamarsi, Ruiming Han, Mohammad Kafi, Stanley Lutts

**Affiliations:** 1Biology Centre, Czech Academy of Sciences, Institute of Plant Molecular Biology, 37005 Ceske Budejovice, Czech Republic; 2Department of Agrotechnology, Faculty of Agriculture, Ferdowsi University of Mashhad, Mashhad 9177948974, Iran; 3School of Environment, Nanjing Normal University, Nanjing 210023, China; 4Groupe de Recherche en Physiologie Vegetale (GRPV), Earth and Life Institute-Agronomy (ELIA), Universite Catholique de Louvain, 5 (Bte 7.07.13) Place Croix du Sud, 1348 Louvain-la-Neuve, Belgium

**Keywords:** *K. pentacarpos*, polymetal, salinity, EDDS, phytoremediation

## Abstract

The ability of plants to accumulate heavy metals is a crucial factor in phytoremediation. This study investigated the effect of NaCl and S,S-ethylenediaminesuccinic acid (EDDS) on heavy metal accumulation in *Kosteletzkya pentacarpos* in soil polluted with arsenic, cadmium, lead, and zinc. The addition of NaCl reduced the bioavailability of arsenic and cadmium, while EDDS increased the bioavailability of arsenic and zinc. The toxicity of the polymetallic pollutants inhibited plant growth and reproduction, but NaCl and EDDS had no significant positive effects. NaCl reduced the accumulation of all heavy metals in the roots, except for arsenic. In contrast, EDDS increased the accumulation of all heavy metals. NaCl reduced the accumulation of arsenic in both the main stem (MS) and lateral branch (LB), along with a decrease in cadmium in the leaves of the main stem (LMS) and zinc in the leaves of the lateral branch (LLB). Conversely, EDDS increased the accumulation of all four heavy metals in the LB, along with an increase in arsenic and cadmium in the LMS and LLB. Salinity significantly decreased the bioaccumulation factor (BF) of all four heavy metals, while EDDS significantly increased it. NaCl had different effects on heavy metals in terms of the translocation factor (TFc), increasing it for cadmium and decreasing it for arsenic and lead, with or without EDDS. EDDS reduced the accumulation of all heavy metals, except for zinc, in the presence of NaCl in polluted soil. The polymetallic pollutants also modified the cell wall constituents. NaCl increased the cellulose content in the MS and LB, whereas EDDS had little impact. In conclusion, salinity and EDDS have different effects on heavy metal bioaccumulation in *K. pentacarpos*, and this species has the potential to be a candidate for phytoremediation in saline environments.

## 1. Introduction

Approximately 830 million hectares of land worldwide, accounting for more than 10% of the world’s total land area, is affected by salinity. Coastal land is one of the most important areas of salt-affected land. However, economic growth in developed areas over the past decades has resulted in the excessive release of heavy metals into coastal waters and land, causing severe heavy metal contamination in these soils [1]. The heavy metal toxicity greatly impairs plant growth and development, inducing a reduction in biomass accumulation and grain yield of cultivated crops. Moreover, heavy metal uptaken by crops is especially risky due to their toxicity to human beings [2].

One of the promising methods to restore heavy metal-contaminated soils is phytoremediation. However, the heavy metal bioavailability in soil is frequently quite low. To increase heavy metal bioavailability in soil, chelating agents such as ethylene diamine tetraacetic acid (EDTA) may be used [3]. EDTA forms complexes with the heavy metal, which allows their detachment from soil particles and subsequent absorption by plant roots. However, EDTA is a persistent agent as it resists degradation by microorganisms. Moreover, due to the high mobility of EDTA–heavy metal complexes, there is a risk of leaching and groundwater contamination if the plant is not able to absorb sufficient amounts of solubilized heavy metals. EDTA may also cause damage to soil bacteria and fungi, as well as to the plant itself. To avoid these major drawbacks, another chelating agent called S, S-ethylenediaminesuccinic acid (EDDS) has been proposed for the remediation of polluted soils. Unlike EDTA, EDDS is biodegradable within 7–32 days and it appears to be quite less damaging to the environment since the risk of leaching heavy metals is reduced [4].

Phytoremediation of heavy metal-contaminated lands, especially coastal saline areas, requires the use of halophyte plant species. *Kosteletzkya pentacarpos* (L.) *Presl.* (formerly designed as *Kosteletzkya virginica*) is a perennial dicot halophyte species of the *Malvaceae* family, which is known for its high salinity tolerance, and it can grow in a wide range of salinity levels. *K. pentacarpos* is able to cope with a high level of salinity in its natural environment (up to 420 mM NaCl), exhibiting a high selectivity for potassium over sodium [5]. It showed the highest growth rates at lower salinity levels around 100 mM NaCl [6]. Polysaccharides play an important role in protecting helophyte species from high salinity by helping to maintain their cellular and physiological functions. Polysaccharides are complex carbohydrates that serve as structural components of the cell wall and as storage molecules for energy [7]. Polysaccharides can help to maintain cell turgor pressure and prevent cellular dehydration, which is especially important in high-salinity environments where water is scarce [8]. It also acts as an osmoprotectant, which means they assist with regulating the concentration of solutes within the plant’s cells. Some helophytes produce specific types of polysaccharides, such as the mucilage produced by the roots of some species. Mucilage could bind and sequester excess salt ions, preventing them from accumulating in the plant’s tissues and causing damage [9]. In *K. pentacarpos*, mucilage increased in shoots, stems, and roots in response to salt stress. It is suggested that the pectic polysaccharide in *K. pentacarpos* could be involved in Na^+^ fixation due to the high proportion of rhamnose and uronic acid in stem mucilage [6]. 

Apart from salt tolerance, *K. pentacarpos* is also able to cope with heavy metal pollution in salt marsh conditions and it could therefore be recommended as an interesting tool for the phytomanagement of heavy metal-polluted coastal areas [10]. Han et al. [11] demonstrated that NaCl differently interfered with Cd and Zn toxicities in this wetland species. Salinity reduced both Cd and Zn accumulation. The distribution of heavy metals among plant organs also appeared differently affected by salinity since Cd was reduced mainly in the leaves while Zn was reduced in the roots. Management of heavy metal oxidative stress by *K. pentacarpos* appeared as a crucial component of resistance to Cd [12] or Zn [11]. However, there are three limitations in these experiments: (1) These data were obtained for plants exposed to one single pollutant (Cd, Zn, and Cu) while this species in field conditions is frequently exposed simultaneously to several pollutants. (2) All of the experiments are in the short term and there is no information reflecting the response of *K. pentacarpos* to the combination of heavy metal toxicity in the long term in relation to the whole life span. (3) The long-term effect of polysaccharides in plants to cope with heavy metal stress is unknown. 

In the present study, we created an contaminated (spiked) soil with a moderate concentration of four heavy metals/metalloids reflecting the high level of pollution encountered by the plants in contaminated salt marshes (arsenic (As), cadmium (Cd), lead (Pd), and zinc (Zn), respectively). In addition, NaCl and EDDS (S, S-ethylene diamine succinic acid) were provided to the soil. Plants were cultivated in the contaminated soil for nearly half a year until they produced fruit. The purpose of the experiment was to compare the behavior of plants growing in polymetallic polluted soil and those maintained on unpolluted soil as well as to specify the impact of salinity and/or EDDS on this response. It is hypothesized that salinity may reduce the metal(oid) absorption and translocation in plants while EDDS may improve it due to the increase in the metal(oid)s’ bioavailability. 

## 2. Results and Discussion

### 2.1. Plant Growth and Reproduction

All tested plants remained alive until the end of treatments. Because the root of each plant was entwined together and fibrous roots were quite developed after 16 weeks of cultivation, it is impossible to obtain accurate results of root dry weight. In this case, the dry weight of the upper parts, stem (main stem and lateral branch), and leaf (leaf on the main stem and on lateral branch) were presented in Figure 1. Except for SPKEA treatment, the MS dry weight and LB dry weight of plants growing in polluted soil significantly decreased (*p* < 0.05, Figure 1a,b), compared to plants growing in non-polluted soil. The LB dry weight of plants in SPK treatment sharply decreased by 86%. However, the polymetallic treatment seemed to have no significant impact on LMS dry weight (*p* > 0.05), except for SPKEA treatment (Figure 1c). In contrast, polymetallic significantly decreased the LLB dry weight in the presence and absence of NaCl or/and EDDS (*p* < 0.05, Figure 1d). On the other hand, NaCl and EDDS had slightly positive impacts on stem and leaf dry weight, suggesting partial polymetallic toxicity alleviation.

From our result (Figure 1), it is suggested that heavy metal toxicity had a more negative impact on the shoot ramification (the branches and leaves in branch development) than the main stem development in *K. pentacarpos*. In Han’s study [10], heavy metals such as cadmium indeed strongly inhibited axillary bud development. Furthermore, the addition of salinity or/and EDDS also had an obvious positive effect on the axillary bud development, especially in the EDDS treatment in the presence or absence of the salinity. It is suggested that EDDS significantly promoted sunflower growth, resulting from Zn and Pb uptake reduction [13]. 

Morphological parameters were measured when plants were transferred to columns and included the stem height, the number of lateral branches, the number of leaves on the main stem (LMS), the number of leaves on the lateral branches (LLB), the total number of flowers, as well as the total number of fruits (Figure 2). The addition of NaCl or/and EDDS had no effect on the morphology of the plant in the presence or absence of heavy metals. After 16 weeks of growth in soil, the stem elongation of *K. pentacarpos* growing in columns with heavy metal-polluted treatment in the presence and absence of NaCl or/and EDDS was slightly but significantly inhibited (*p* < 0.05). The number of LNs exhibited a similar tendency as a consequence of heavy metal toxicities (Figure 2b). After 8 weeks, the number of LMSs and the number of LLNs were significantly different between non-polluted and polluted treatments (Figure 2c,d). At 16 weeks, the number of LMSs and LLBs of plants growing in polluted soil significantly decreased (*p* < 0.05), especially the number of LLNs, which decreased by 68%. In the present study, a drastic effect of poly-metals on lateral branch growth, especially in the number of LLN, was observed, which demonstrated once again that heavy metals deeply affect lateral ramification.

For reproductive parameters, the plant had more flowers and fruits growing in non-polluted soil in the last record (Figure 2e,f). An interesting observation was that plants growing in polluted soil initiated the reproductive stage at least 10 days earlier than plants growing in non-polluted soil. The first flower bud and fruit appeared in plants growing in polluted soil. Ryser and Sauder [14] reported that even if the concentration of heavy metals was very low, the flowering phenology of *Hieracium pilosella* was still very sensitive to metals. It delayed and reduced *Hieracium pilosella* reproduction. In contrast, it seems the opposite in *K. pentacarpos*, in which the explanation may be that plants exposed to heavy metals tried to enter the flowering phase in advance in order to use the available strategy for precocious reproduction rather than vegetative growth. 

### 2.2. Heavy Metal Concentration and Bioavailability in Soil 

All heavy metals were detected in all treatments. The As, Cd, Pb and Zn contents in the non-polluted soil were lower than the threshold values for agricultural uses (30, 1.0, 200, and 155 mg/kg, respectively). In our results, the total concentration of all heavy metals and As was higher in EDDS treatment than without EDDS in polluted soil (Table 1). In terms of the EDDS function, it is well known that the EDDS application could increase the soil heavy metals’ bioavailability, leading to promoting heavy metal uptake by the plant, and eventually improving phytoextraction efficiency. However, the total concentration of heavy metals and As decreased in the absence of EDDS, which suggests that EDDS did not have a positive impact on heavy metal and As uptake in *K. pentacarpos*. In contrast, the concentration of heavy metals and As was lower in the presence of salt than in the absence because the total concentrations and total amount of pollutants were fixed in this experiment. It could be explained that the reduction part of heavy metals and As was transferred to the plants’ tissue. It is suggested by Weggler-Beaton [15] that cadmium concentrations in soil solution and shoots of wheat (*Triticum aestivum* cv. *Halberd*) and Swiss chard (*Beta vulgaris* cv. *Foodhook Giant*) plants increased linearly with increasing Cl concentration in the soil solution of the biosolids-amended soil. The additional Cl- could form a CdCl^+^ complex with Cd^2+^, in which, the activity of the CdCl^+^ complex correlated best with the Cd uptake of both plant species. However, Manousaki et al. [16] pointed out that although the increasing salinity increased cadmium uptake by *A. halimus* L., in the case of lead, there was not a clear effect of the presence of salt on the lead accumulation in the plant tissues.

The bioavailable fractions of pollutants in the soil have also been determined (Table 2). Only polluted soil values are above the detection limit (LD) except for Pb which remained extremely low. The percentage of the corresponding total fraction is shown in Table 2. EDDS increased the bioavailability of As and Zn, while NaCl decreased the bioavailability of As and Cd in polluted soil.

### 2.3. Heavy Metal Concentration in the Vegetative Organs 

All heavy metals were detected in the roots of plants growing on non-polluted and polluted treatments (Table 3). The highest concentration of As in the root was found in SPKEA. Both EDDS and salinity increased the accumulation of As in roots exposed to polluted soil. NaCl induced the strongest increase by 181% and 139% in the presence and absence of EDDS, respectively. The concentrations of Pb in non-polluted treatment were always lower than 2 mg/kg. In polluted soil, EDDS significantly increased Pb accumulation in the root by 30% in the absence of NaCl. The addition of salinity had no significant effect on the Pb levels in the roots. The concentrations of Cd in the roots of plants grown on unpolluted soils were lower than or equal to 0.9 mg/kg. EDDS induced a significant increase in Cd concentration in roots in polluted soil (*p* < 0.05). In contrast, salinity significantly decreased the Cd accumulation in the root (*p* < 0.05). The highest concentration of Zn was found in SPKE (99.2 ± 1.2 mg/kg). The addition of NaCl significantly decreased the accumulation of Zn in roots growing in polluted soil (*p* < 0.05) while EDDS significantly increased the level of Zn in the absence of NaCl (*p* < 0.05). 

In summary, EDDS enhanced the concentration of all pollutants in the root of *K. pentacarpos* growing in polluted soil. In contrast, NaCl reduced the accumulation of all pollutants in the root; only As increased, although its bioavailability in the presence of NaCl in polluted soils decreased. It has been widely reported [17,18] that the additional salinity could enhance plant resistance to As resulting from decreasing As uptake in the root. However, our study showed the opposite result, which could explain that the relatively low concentration of As was not toxic for *K. pentacarpos*. At the same time, in long-term cultivation, the additional salinity increased the plant’s tolerance to As toxicity.

All heavy metals were detected in the main stem (MS) and lateral branches (LB) of plants (Table 4). As concentration in MS and LB remained lower than the limit of quantification (LQ, 8 ppb). The highest concentration of As was found in SPKE in LB. In polluted soil, NaCl significantly decreased the concentration of As in both MS and LB. The addition of EDDS reduced the As concentration in MS while it increased As in LB when plants were grown in polluted soil. The stem concentrations of Pb in the non-polluted treatment were always lower than 2 ppb. The addition of NaCl significantly decreased the Pb level in the LB of plants grown on polluted soil except in the absence of EDDS (*p* < 0.05). EDDS had a limited impact on Pb accumulation in the shoot. The concentrations of Cd found in the MS and LB of plants grown on unpolluted soils are all less than LQ (0.8 mg/kg DM). 

EDDS significantly decreased the Cd concentration in MS in the presence of NaCl (*p* < 0.05) in polluted soil while it increased it in LB in the absence of salt (*p* < 0.05). On the other hand, the additional NaCl significantly increased the Cd concentration in MS in the absence of EDDS (*p* < 0.05) while it decreased it in LB in the presence of EDDS (*p* < 0.05). For zinc, the addition of NaCl significantly decreased the accumulation of Zn in LB in polluted soil (*p* < 0.05) and had no impact on it in MS. EDDS significantly increased the Zn in the presence and absence of NaCl in MS as well as in LB (*p* < 0.05). In summary, NaCl reduced As accumulation in both MS and LB, as well as Pb and Zn accumulation in LB. Conversely, EDDS increased the accumulation of all four heavy metals in LB and Zn in MS. 

The concentration of As in LMS and LLB from plants growing in non-polluted soil was far lower than for plants growing on the polluted substrate (Table 5). In polluted treatment, NaCl significantly decreased the concentration of As in the presence of EDDS in LLB. In contrast, EDDS increased the As concentration in LMS as well as LLB. NaCl significantly decreased the Pb content in the LMS and LLB of plants grown in polluted soil in the absence of EDDS (*p* < 0.05). EDDS also significantly decreased Pb concentration in the presence or absence of NaCl in LLB (*p* < 0.05). NaCl significantly decreased the Cd concentration in LMS (*p* < 0.05) but had no impact on LLB. EDDS significantly increased Cd in LMS and LLB in the presence or absence of NaCl in polluted soil (*p* < 0.05). NaCl significantly decreased Zn accumulation in LMS and LLB in polluted soil (*p* < 0.05). EDDS significantly decreased the level of Zn in LMS in the presence of NaCl (*p* < 0.05) but had no impact on Zn in LLB.

In our result, the salinity application differently affected different pollutant species. Unlike our results, Vromman [18] indicated that NaCl increased As translocation from the root to the shoot of Atriplex atacamensis Phil. but had no impact on As distribution between the apoplasm and the symplasm. For lead, the salinity decreased its translocation to LMB and LMB. Li demonstrated that the additional salinity induced growth stimulation to dilute Pb in *Suaeda salsa* [19]. In the case of cadmium, although the salinity did not obviously affect the concentration of Cd in *K. pentacarpos*, the amount of Cd in plants increased, which has similar results to Ghnaya’s research on *Sesuvium portulacastrum* [20]. 

The EDDS application had limited impact on As accumulation in plant shoots, while it increased Cd concentration in LMS and LB in our study. Lan indicated that the biodegradable EDDS had the potential for enhancing the efficiency of remediation by *S. orientalis* in Cd-polluted soil as a consequence of Cd accumulation enhancement in plant tissue [21]. For lead, EDDS differently affects its accumulation in the shoot in different plant species. It is also suggested that the application of EDDS had less impact on Pb accumulation (only 310 mg kg^−1^) in the shoot of *Cynara cardunculus* than it in EDTA treatment (1332 mg kg^−1^ DW) [22]. However, in Attinti’s study [23], the concentration of lead in shoots of vetiver grass increased dramatically after EDDS application, indicating that chelation occurred. In our study, the additional EDDS strongly decreased Pb concentration in leaves in *K. pentacarpos*. In Marques’ study [24], the EDDS application enhanced the accumulation in leaves, stems, and roots of *Solanum nigrum* L. grown in Zn-contaminated soil up to 140, 124, and 104%. The same results were recorded for Zn in our study.

The addition of salinity significantly decreased the bioaccumulation factor (BF) of all pollutants in polluted soil, except for Cd in the absence of EDDS (Table 6). On the other hand, EDDS significantly increased the BF value of As, Cd, and Zn but decreased it for Pb. Together with the results in the total concentration of all pollutants in contaminated soil, EDDS reduced Pb accumulation in *K. pentacarpos* (total Pb concentration of Pb in non-EDDS treated soil decreased). In terms of the translocation factor (TFc), NaCl increased the TFc of Cd and decreased it for As and Pb in the presence or absence of EDDS. Furthermore, the addition of EDDS reduced TFc for all heavy metals except Zn in the presence of NaCl in polluted soil.

In a global view, the salinity or EDDS differently affects pollutant accumulation in *K. pentacarpos* depending on the pollutant species. Furthermore, from the result in Table 5, Table 6 and Table 7, it is suggested that these two applications had opposite influences on pollutant accumulation in *K. pentacarpos*, which is also confirmed in most results from EDDS + salinity treatment (the value in EDDS + salinity treatment is always between the EDDS treatment and the salinity treatment). It is easy to understand that in general, the EDDS application should increase heavy metal bioavailability in soil. More ions are released and then absorbed by plants. In contrast, it is widely studied that the salinity application always alleviates pollutants’ toxicity in halophyte plant species, with one of the most important reasons that it could reduce pollutant accumulation in plant tissues. 

### 2.4. Structural Polysaccharides Analysis

Polymetallic pollution induced a slight modification in the cell wall composition (Table 7). In the main stem, the content of lignin and hemicellulose in plants growing in polluted soil in the presence and absence of NaCl or/and EDDS decreased. Similarly, the content of hemicellulose in the lateral branch of plants growing in polluted soil in the presence and absence of NaCl or/and EDDS also decreased significantly albeit slightly. Secondly, NaCl increased the cellulose content in the main stem and lateral branches when plants grew in the presence and absence of EDDS in polluted soil. 

EDDS had little impact on the content of lignin, hemicellulose, as well as cellulose. Due to the electrical neutrality and the relative stability, the cellulose as well as the lignin had a limited effect on metal(oid)s fixation, while the hemicellulose had the ability to absorb and chelate pollutants. The synthesis of hemicellulose may be stimulated by some kinds of messenger molecular or plant hormones when plants are stressed by pollutants. Xiong demonstrated that exogenous nitric oxide enhances cadmium tolerance as well as Cd accumulation in rice by increasing pectin and hemicellulose contents in the root cell wall [25]. Moreover, Zhu reported that exogenous auxin alleviates cadmium toxicity in *Arabidopsis thaliana* by stimulating the synthesis of hemicellulose 1 and increasing the cadmium fixation capacity of root cell walls [26]. 

## 3. Materials and Methods

### 3.1. Culture Condition and Plant Material

The soil used for the experiments had a silty texture and came from an agricultural plot located in Louvain-la-Neuve (Avenue Baudouin I) (Appendix A). An aqueous solution containing 4 heavy metals/metalloids was sprayed on a thin layer of soil for a total amount of 700 kg soil. The levels of pollution reached were 155.5, 6.5, 312.5, and 187.5 mg/kg for As, Cd, Zn, and Pb, respectively. The salts used to prepare the solution were As_2_O_3_, CdCl_2_, PbCl_2_, and ZnCl_2_.

Each column (50 cm depth) received 20 kg soil with pH 8.20 ± 0.01, and the average organic matter was 13.96 g/kg soil. Eight treatments were then considered with 3 columns per treatment, as detailed in Table 8. For salt treatments, 60 g NaCl (3 g/kg soil) was added, whereas for EDDS treatment, 16 g of EDDS molecule (0.8 g/kg of soil) was used. All chemicals were purchased from SigmaChemical Belgium.

The germination of *K. pentacarpos* was performed in trays filled with a perlite and vermiculite mix (1:3 *v*/*v*) and moistened regularly with water from 20 February 2019. Seedlings were grown in a phytotron under a 12 h photoperiod. Eighteen days after sowing, 5 seedlings were transferred from each column into a greenhouse on 10 March 2017. Natural light was supplemented by Philips lamps. After five months (first September 2019), all plants were harvested. The main stem and lateral branches were separately collected as well as leaves on the main stem and on the lateral branches. 

### 3.2. Growth and Reproduction Assessment

Stem height, the number of lateral branches (LBs), leaf number on the main stem (LMS), and the number of leaves on LBs (LLB) were recorded every two weeks (in total, 9 times during the whole experiment) from 25 March 2019 until the end of the study. The first flower and fruit appeared on 5 June and 21 June, respectively; therefore, from that time, the number of flower buds (NF), as well as fruits (NFR), were counted every 10 days. 

Leaves in the main stem and lateral branch were harvested, respectively, and immediately frozen in liquid nitrogen. Fresh material from the main stem, lateral branches, leaves on the main stem, and leaves on lateral branches from the remaining 8 plants were weighed and then incubated in an oven for 72 h at 70 °C for dry weight determination. Meanwhile, the soil in the upper, middle, and bottom levels was collected and pooled from each column as one treatment. 

### 3.3. Evaluation of Ion Concentration in Plants 

Dried samples were ground to a fine powder using a porcelain mortar and a pestle, digested in 35% HNO_3_ and evaporated to dryness on a sand bath at 80 °C [12]. The minerals were incubated with a mix of 37% HCl and 68% HNO_3_ (3:1), and the mixture was slightly evaporated. The minerals were finally dissolved in HCl 0.1 N. The ion concentrations were determined by SOLAAR S4 atomic absorption spectrometry (Thermo Scientific, Cambridge, UK). 

The translocation factor was estimated on the basis of ion concentration (TFc) according to
TFc = *C_shoot_*/*C_root_*

The bioaccumulation factor (BF) calculation is according to:BF = *C_shoot_*/*C_soil_*

### 3.4. Ion Concentration and Bioavailability in Soil

Followed by Lambrecht [27], a sample of approximately 0.3 g of mixed soil (collected from 5 cm and 20 cm depths) was weighed and digested with 10 mL of HCl on an electric hot plate at 190 °C until the solution was reduced to 3 mL. Then, 5 mL of HF (40%, *w*/*w*), 5 mL of HNO_3_ (63%, *w*/*w*), and 3 mL of HClO_4_ (70%, *w*/*w*) was added, and the solution was digested until no black material remained. The digestion was continued further with 3 mL of HNO_3_, 3 mL of HF, and 1 mL of HClO_4_ until the silicate minerals had completely disappeared. Finally, the digestion solution was transferred to a 25 mL volumetric polypropylene tube, and 1% HNO_3_ was added to bring the sample up to a fixed volume for the metal determinations. After filtering the digested samples through a syringe filter (0.45 μm), the concentrations of heavy metals were measured by using inductively coupled plasma mass spectrometry (ICP-MS, Thermo Electron Corporation, Waltham, MA, USA).

For heavy metal bioavailability analysis in soil, samples were extracted by a CaCl_2_ solution: 1 g dry soil was weighed into 15 falcon tubes and 10 mL 0.01 M CaCl_2_ was added to each sample. After 24 h of incubation at room temperature, the samples were centrifuged for 20 min at 3000× *g*. The filtered supernatant was added to 20 μL HNO_3_. The concentrations of bioavailable heavy metals were measured by using inductively coupled plasma mass spectrometry (ICP-MS, Thermo Electron Corporation). 

Heavy metal bioavailability = concentration of bioavailable heavy metal (mg kg^−1^ soil) /total concentration of heavy metal (mg kg^−1^ soil).

### 3.5. Structural Polysaccharides Analysis by the Van Soest Method

The structural polysaccharides were determined by the detergent fiber assay which is based on the Van Soest (VST) method [28,29]. Briefly, the neutral detergent fiber residue (NDF) was determined by the use of extraction 1:0.1 mmol/L phosphate buffer at pH 7 for 15 min at 90 °C and the addition of an analytical thermostable α-amylase for samples that contained starch; extraction 2: Van Soest neutral detergent at 100 °C for 1 h and the addition of sodium sulfite. The acid detergent fiber residue (ADF) was determined by two successive extractions. The first extraction was performed with the Van Soest neutral detergent, as described above, without the addition of sodium to the Van Soest neutral detergent. Then, it was followed by the extraction with the Van Soest acid detergent at 100 °C for 1 h. The acid detergent lignin residue (ADL) was determined from the acid detergent fiber residue by extracting it with sulfuric acid 12.2 mol/L at room temperature for 3 h. Sodium sulfite was added to the Van Soest neutral detergent extraction, and it was not added to the Van Soest neutral detergent extraction prior to the ADF extractions because it was a standard recommendation. The cellulose VST, hemicelluloses VST, and lignin VST contents are calculated as the difference between ADF and ADL, the difference between NDF and ADF, and ADL, respectively, and are expressed in g 100 g^−1^ OM (organic matter).

### 3.6. Statistical Analysis

Parts of tissue materials were dried in the 70 °C oven, which were used to analyze growth parameters, ionic determination, and structural polysaccharide analysis. The data obtained were subjected to analysis of variance, one-way ANOVA and two-way ANOVA (treatment and duration of stress as the level of classification), using SPSS software. Each value of the mean in dry weight, metal(oid) concentrations of different tissues, and structural polysaccharides had 3 biological replicates, respectively. Each growth parameter had 15 biological replicates. The concentration of metal(oid)s in the mixed soil sample was tested once. The statistical significance of the results was analyzed by the Student–Newman–Keuls test at the 5% level (*p* < 0.05).

## 4. Conclusions

*K. pentacarpos* is not regarded as a hyperaccumulating plant species since it is not able to accumulate pollutants in plant tissue up to the hyperaccumulator level. At the same time, salinity reduces heavy metal absorption, translocation, and accumulation. However, salinity improves the capacity of the plant to cope with a mixture of heavy metals with specific plant physiological properties sustaining heavy metal resistance. 

A chelating agent (EDDS) was tested in order to mitigate the impact of NaCl on heavy metal bioavailability; it clearly increased the bioavailability of As and Zn while NaCl decreased the bioavailability of As and Cd on polluted soil. Data are not so clear regarding the impact of EDDS on heavy metal accumulation in plants: it is supposed to increase as a result of a chelate-induced increase in bioavailability, but no uniform trend was reported in this respect. Accumulation varies between roots, stems, and leaves but also depends on the location of the shoot (main stem versus lateral branches) thus confirming that the plant behavior is not uniform and that long-term experiments need to integrate the impact of plant morphogenesis in relation to heavy metal distribution. Considering the toxicity of EDTA, EDDS was chosen as a non-toxic chelating agent, but we apply one single dose of EDDS at the beginning of the experiment at the time of spiked-soil contamination, and it cannot be excluded that this biodegradable EDDS was not efficient anymore after 16 weeks of treatment.

## Figures and Tables

**Figure 1 plants-12-01656-f001:**
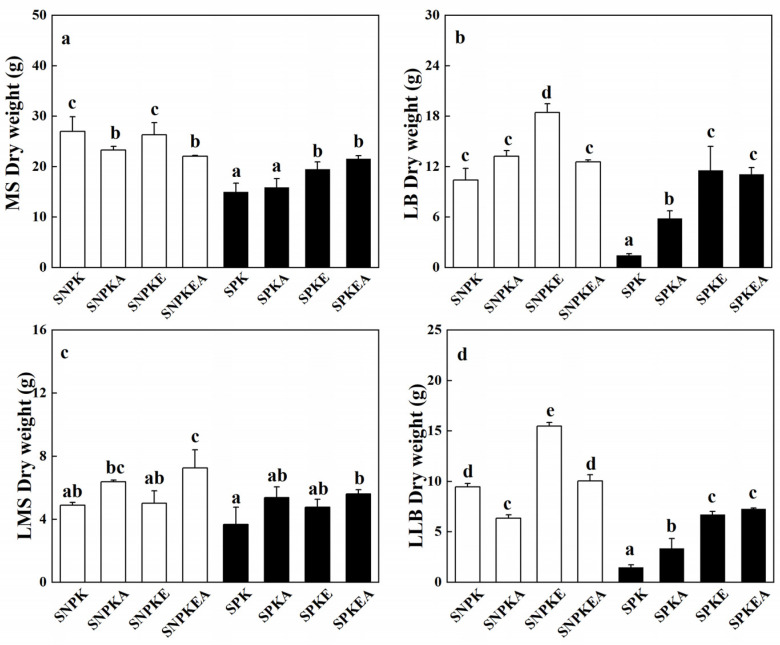
Dry weight of MS (main stem, (**a**)), LB (lateral branch, (**b**)), LMS (leaf on main stem, (**c**)) and LLB (leaf on lateral branch, (**d**)) in *K. pentacarpos* cultivated in non-polluted or polluted soil for five months in the presence or absence of NaCl or/and EDDS. Each value is the mean of 3 replicates and vertical bars are S.E. Values exhibiting different letters are significantly different at *p* < 0.05 according to SNK test.

**Figure 2 plants-12-01656-f002:**
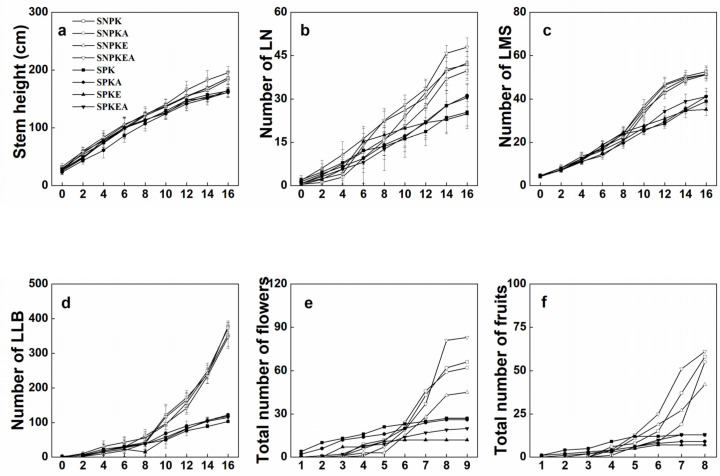
Growth parameters of *K. pentacarpos* in non-polluted or polluted soil during five months in the presence or absence of NaCl or/and EDDS. The stem height (**a**), number of lateral branch (LB, (**b**)), number of leaves on main stem (LMS, (**c**)), and number of leaves on lateral branch (LLB, (**d**)) were recorded every two weeks until the 16th week (**a**–**d**). Total number of flowers (**e**) and fruits (**f**) were recorded every ten days from the first day that they appeared, in total, 9 and 8 times, respectively. Each value is the mean of 15 replicates and vertical bars are S.E. in Figure 1a–d.

**Table 1 plants-12-01656-t001:** Heavy metal (As, Cd Pb, and Zn) concentration (mg kg^−1^ soil) in different layers of mixed soil after cultivation of *K. pentacarpos* in the presence and absence of NaCl or/and EDDS for 5 months. The sample was tested once.

Concentration (mg kg^−1^ Soil)	As	Cd	Pb	Zn
SNPK	5.4	0.92	22.1	83.2
SNPKA	8.2	0.28	24.2	73.7
SNPKE	9.6	0.41	44	67.2
SNPKEA	7.6	0.28	23.4	67.6
SPK	173	6.1	347	216
SPKA	127	4.7	230	184
SPKE	179	6.2	360	218
SPKEA	146	5.5	253	215

**Table 2 plants-12-01656-t002:** The percentage of the bioavailable fraction (CaCl_2_ extraction) of heavy metal (As, Cd, Pb, and Zn) in the total fraction in different layers of mixed soil after cultivation of *K. pentacarpos* in the presence and absence of NaCl or/and EDDS for 5 months. The sample was tested once.

Bioavailability (%)	As	Pb	Cd	Zn
SPK	0.47	<0.25	0.53	0.078
SPKA	0.39	<0.25	0.25	0.088
SPKE	0.69	<0.25	0.46	0.16
SPKEA	0.42	<0.25	0.42	0.16

**Table 3 plants-12-01656-t003:** Total concentration (mg kg^−1^ DW) of heavy metal (As, Pb, Cd, and Zn) in the root of *K. pentacarpos* exposed to non-polluted or polluted soil in the presence or in the absence of NaCl or/and EDDS for five months. Each value is the mean of 3 replicates and vertical bars are S.E. Values exhibiting different letters are significantly different at *p* < 0.05 according to SNK test.

Treatment	As	Pb	Cd	Zn
SNPK	2.5 ± 0.8 a	1.7 ± 0.4 a	0.60 ± 0.09 a	20.6 ± 1.3 a
SNPKA	1.7 ± 0.2 a	1.8 ± 0.2 a	0.90 ± 0.16 a	28.0 ± 5.6 a
SNPKE	2.0 ± 0.5 a	1.5 ± 0.3 a	0.55 ± 0.07 a	25.0 ± 4.8 a
SNPKEA	2.1 ± 0.2 a	1.9 ± 1.1 a	0.73 ± 0.21 a	21.4 ± 0.8 a
SPK	43.0 ± 4.6 b	132 ± 20 b	6.5 ± 0.3 c	79.0 ± 6.7 c
SPKA	121 ± 4.4 d	126 ± 12 b	4.3 ± 0.8 b	66.2 ± 2.0 b
SPKE	53.9 ± 7.4 c	171 ± 16 c	10.4 ± 0.7 d	99.2 ± 1.2 d
SPKEA	129 ± 2.4 e	153 ± 27 bc	7.2 ± 2.9 c	72.4 ± 10.1 bc

**Table 4 plants-12-01656-t004:** Total concentration (mg kg^−1^ DW) of heavy metal (As, Pb, Cd, and Zn) in stem (MS: main stem; LB: lateral branch) of *K. pentacarpos* exposed to non-polluted or polluted soil in the presence or in the absence of NaCl or/and EDDS for five months. Each value is the mean of 3 replicates and vertical bars are S.E. Values exhibiting different letters are significantly different at *p* < 0.05 according to SNK test.

Treatment	As (mg kg^−1^ DW)		Pb (mg kg^−1^ DW)		Cd (mg kg^−1^ DW)		Zn (mg kg^−1^ DW)	
	MS	LB	MS	LB	MS	LB	MS	LB
SNPK	0.63 ± 0.12 a	0.59 ± 0.08 a	0.25 ± 0.02 a	0.33 ± 0.05 a	0.34 ± 0.04 a	0.46 ± 0.09 a	13 ± 2.9 a	21 ± 1.8 a
SNPKA	0.56 ± 0.12 a	0.63 ± 0.11 a	0.12 ± 0.02 a	0.28 ± 0.11 a	0.52 ± 0.1 a	0.43 ± 0.07 a	17 ± 1.0 abc	22 ± 0.47 a
SNPKE	0.53 ± 0.16 a	0.59 ± 0.15 a	0.20 ± 0.02 a	0.36 ± 0.15 a	0.29 ± 0.03 a	0.46 ± 0.04 a	14 ± 0.6 ab	22 ± 2.0 a
SNPKEA	0.53 ± 0.12 a	0.57 ± 0.15 a	0.19 ± 0.08 a	0.25 ± 0.08 a	0.62 ± 0.12 a	0.63 ± 0.19 a	17 ± 1.4 abc	23 ± 2.2 a
SPK	2.5 ± 0.2 d	2.2 ± 0.17 d	5.7 ± 0.2 c	5.4 ± 0.8 d	2.0 ± 0.2 b	3.7 ± 0.15 b	22 ± 5.6 cd	45 ± 2.6 c
SPKA	1.8 ± 0.3 c	1.1 ± 0.02 b	5.8 ± 0.4 c	2.6 ± 0.1 b	2.7 ± 0.1 c	3.5 ± 0.4 b	20 ± 1.7 bc	37 ± 1.7 b
SPKE	2.3 ± 0.3 d	2.8 ± 0.3 e	6.6 ± 0.5 d	5.9 ± 0.6 d	2.2 ± 0.5 b	5.2 ± 0.2 c	30 ± 4.0 e	62 ± 4.3 e
SPKEA	1.0 ± 0.2 b	1.7 ± 0.3 c	4.7 ± 0.2 b	4.0 ± 0.5 c	2.1 ± 0.1 b	3.6 ± 0.4 b	27 ± 1.4 de	52 ± 3.4 d

**Table 5 plants-12-01656-t005:** Total concentration (mg kg^−1^ DW) of heavy metal (As, Pb, Cd, and Zn) in leaf (LMS: leaf on main stem; LLB: leaf on lateral branch) of *K. pentacarpos* exposed to non-polluted or polluted soil in the presence or in the absence of NaCl or/and EDDS for five months. Each value is the mean of 3 replicates and vertical bars are S.E. Values exhibiting different letters are significantly different at *p* < 0.05 according to SNK test.

Treatment	As Concentration (mg/kg DW)		Pb Concentration (mg/kg DW)		Cd Concentration (mg/kg DW)		Zn Concentration (mg/kg DW)	
v	LMS	LLB	LMS	LLB	LMS	LLB	LMS	LLB
SNPK	0.46 ± 0.14 a	0.40 ± 0.09 a	1.1 ± 0.1 a	1.0 ± 0.2 a	1.2 ± 0.4 a	0.95 ± 0.11 a	44 ± 10.9 a	48 ± 6.3 a
SNPKA	0.39 ± 0.07 a	0.27 ± 0.04 a	1.0 ± 0.1 a	1.3 ± 0.2 a	1.8 ± 0.01 a	1.6 ± 0.3 a	55 ± 6.3 a	55 ± 3.7 a
SNPKE	0.41 ± 0.04 a	0.41 ± 0.07 a	1.0 ± 0.2 a	1.4 ± 0.2 a	0.92 ± 0.1 a	0.88 ± 0.05 a	47 ± 8.3 a	51 ± 6.9 a
SNPKEA	0.46 ± 0.04 a	0.33 ± 0.08 a	1.0 ± 0.1 a	1.2 ± 0.2 a	1.5 ± 0.2 a	1.6 ± 0.3 a	67 ± 1.8 b	60 ± 6.7 a
SPK	2.3 ± 0.3 b	3.7 ± 0.6 bc	9.4 ± 2.1 c	10.6 ± 1.9 c	5.3 ± 0.5 c	5.8 ± 0.6 bc	101 ± 3.2 d	115 ± 12 c
SPKA	2.0 ± 0.3 b	3.2 ± 0.2 b	4.4 ± 1.0 b	6.3 ± 0.6 b	4.3 ± 0.5 b	5.0 ± 0.2 b	92 ± 4.1 d	78 ± 13 b
SPKE	2.8 ± 0.3 c	5.1 ± 0.4 d	5.7 ± 1.5 b	3.7 ± 0.9 a	7.2 ± 1.2 d	6.4 ± 0.8 c	101 ± 1.4 d	102 ± 9.1 c
SPKEA	2.8 ± 0.1 c	4.0 ± 0.6 c	7.4 ± 1.0 bc	3.6 ± 0.4 a	5.7 ± 0.5 c	6.4 ± 0.9 c	81 ± 2.7 c	78 ± 1.5 b

**Table 6 plants-12-01656-t006:** Bioaccumulation factor (BF) and translocation factor which is estimated on the basis of concentration (TFc) of *K. pentacarpos* seedlings grown in polluted soil in the presence or absence of NaCl or/and EDDS for five months. Each value is the mean of 3 replicates and vertical bars are S.E. Values exhibiting different letters are significantly different at *p* < 0.05 according to SNK test.

	As		Pb		Cd		Zn	
	BF	TF_C_	BF	TF_C_	BF	TF_C_	BF	TF_C_
SPK	1.7 ± 0.01 b	6.0 ± 0.04 d	2.1 ± 0.04 d	5.0 ± 0.10 d	61 ± 2.0 a	49 ± 1.6 b	26 ± 1.4 b	61 ± 3.2 a
SPKA	1.2 ± 0.01 a	1.5 ± 0.01 b	1.7 ± 0.01 b	4.3 ± 0.01 c	61 ± 0.24 a	75 ± 0.30 c	22 ± 0.21 a	61 ± 0.58 a
SPKE	1.9 ± 0.02 c	5.5 ± 0.06 c	1.9 ± 0.01 c	3.4 ± 0.02 b	81 ± 0.63 c	41 ± 0.32 a	31 ± 0.58 c	58 ± 1.1 a
SPKEA	1.2 ± 0.31 a	1.4 ± 0.02 a	1.5 ± 0.01 a	3.1 ± 0.02 a	69 ± 0.71 b	50 ± 0.52 b	25 ± 0.24 b	66 ± 3.2 b

**Table 7 plants-12-01656-t007:** Contents of lignin, hemicellulose, and cellulose in the main stem and lateral branches of *K. pentacarpos* seedlings grown in polluted soil in the presence or absence of NaCl or/and EDDS for five months. Each value is the mean of 3 replicates and vertical bars are S.E. Values exhibiting different letters are significantly different at *p* < 0.05 according to SNK test.

Treatment	Main Stem (MS)			Lateral Branch (LB)		
	Lignin (g/100 g OM)	Hemicellulose (g/100 g OM)	Cellulose (g/100 g OM)	Lignin (g/100 g OM)	Hemicellulose (g/100 g OM)	Cellulose (g/100 g OM)
SNPK	15.2 ± 0.84 cd	10.9 ± 0.48 b	51.0 ± 0.19 bc	11.3 ± 1.2 a	10.9 ± 0.41 a	44.9 ± 1.1 bc
SNPKA	14.5 ± 0.50 acd	11.6 ± 0.41 bc	52.1 ± 3.5 c	10.6 ± 0.75 a	11.6 ± 0.25 b	43.8 ± 0.74 ac
SNPKE	15.2 ± 0.56 cd	12.7 ± 1.1 c	50.3 ± 0.38 ac	11.6 ± 0.66 ab	11.7 ± 0.31 b	44.6 ± 1.6 bc
SNPKEA	15.4 ± 0.84 cd	12.8 ± 1.2 c	52.4 ± 0.27 c	10.7 ± 0.28 a	13.2 ± 0.12 d	44.5 ± 0.64 bc
SPK	13.9 ± 0.12 ab	9.5 ± 0.19 a	48.9 ± 0.19 ab	13.7 ± 1.5 bc	10.4 ± 0.63 a	42.3 ± 0.08 a
SPKA	14.7 ± 0.46 bd	10.6 ± 0.07 ab	55.8 ± 0.47 d	11.0 ± 2.4 a	10.8 ± 0.20 a	46.8 ± 1.6 d
SPKE	13.4 ± 0.67 a	11.0 ± 0.66 b	48.4 ± 0.53 a	11.8 ± 0.94 ac	10.4 ± 0.26 a	43.1 ± 0.19 ab
SPKEA	14.6 ± 0.80 bc	11.5 ± 0.59 b	52.0 ± 0.67 c	11.1 ± 0.70 a	12.3 ± 0.42 c	46.9 ± 1.5 d

**Table 8 plants-12-01656-t008:** Code and treatment of the experiment on *K. pentacarpos* subjected to pollution of heavy metal and the presence and absence of EDDS or/and NaCl.

Code	Treatment
SNPK	No-polluted soil + plant
SNPKA	No-polluted soil + plant + NaCl
SNPKE	No-polluted soil + plant + EDDS
SNPKEA	No-polluted soil + plant + NaCl + EDDS
SPK	Polluted soil + plant
SPKA	Polluted soil + plant + NaCl
SPKE	Polluted soil + plant + EDDS
SPKEA	Polluted soil + plant + NaCl + EDDS

## Data Availability

All available data are contained within the article.

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
