# Peer review of "Effect of NaCl and EDDS on Heavy Metal Accumulation in Kosteletzkya pentacarpos in Polymetallic Polluted Soil"

_plants, 2023, doi:10.3390/plants12081656_

Round 1

Reviewer 1 Report

The manuscript "Effect of NaCl and EDDS on heavy metal accumulation in Kosteletzkya pentacarpos in polymetallic polluted soil" by M. Zhou et al. The general topic of the manuscript covers aims and scope the journal of PLANTS. The study investigates the model experiment to compare the behavior of plants of Kosteletzkya pentacarpos growing on polymetallic polluted and unpolluted soil as well as to evaluate the effect of salinity (NaCl) and EDDS on this response. The Introduction section is very informative and well represents the importance of this study.

Results presented in manuscript are interesting, however, the paper needs to be minor revised before publication.

Some comments are below.

1. In section 2.1, give the physical and chemical properties (Corg, pH, CaCO3, content of particle size, etc.) of the original (unpolluted) soil that was used for the model experiment.

2. Why were used the chloride forms of the metals (CdCl2, PbCl2 and ZnCl2) rather than the acetate or nitrate forms?

3. How many field replicates were used in the experiment for each variant?

4. When polluted with chlorides, how was removed the accompanying Cl-anion from the soil? Was this factor taken into account?

5. In sections 2.3 and 2.4, provide links in text and in reference list to the methods by which the acid decomposition of plants and soils was carried out.

6. In Figure 2 (a-f), give the names for the horizontal axes.

Author Response

.

Reviewer 2 Report

Dear Authors,

This manuscript is very interesting. The authors put a lot of effort into planning and carrying out the research. They drew interesting conclusions. In my opinion, there is chaos in the description of the results. I also believe that the discussion of the results is minimal and this part of the manuscript should be expanded. It is worth thinking about redrafting the Results and Discussion chapter. Authors should formulate a clear research goal. I wonder if their goal was to enhance phytoremediation by using EDDS or to mitigate heavy metal bioavailability by using NaCl?

In addition, the manuscript should be prepared in accordance with the guidelines for authors (especially the list of literature and its citation in the text). Unfortunately, this is not currently the case.

Please write equations using the appropriate MS Office function.

I have attached a document in which I marked in green the places that need to be corrected.

Author Response

.
